# Deep learning-based near-field effect correction method for Controlled Source Electromagnetic Method and application

**Wei Luo[1,2,3], Xianjie Chen[2,3]\*, Shixing Wang[1], Siwei Zhao[1], Xiaokang Yin[1], Xing Lan[3], Peifan Jiang[2], Shaojun Wang[3]**

**1** China Railway Eryuan Engineering Group CO., LTD, Chengdu, China, **2** Chengdu University of Technology, Chengdu, China, **3** Sichuan Natural Resources Investigation and Design Group CO., LTD, Chengdu, China

\* cxianjie@foxmail.com

**Data Availability Statement:** All data files are available from the Harvard Dataverse(https://doi.org/10.7910/DVN/OMJNQJ) The code is available

## Abstract

Addressing the impact of near-field effects in the Controlled Source Electromagnetic Method(CSEM) has long been a focal point in the realm of geophysical exploration. Therefore, we propose a deep learning-based near-field correction method for controlled-source electromagnetic methods. Initially, diverse datasets for a layered geologic model are generated through forward simulation. Building upon the characteristics of near-field effects, a deep learning network utilizing LSTM-CNN is meticulously constructed. Multiple experiments are executed to scrutinize the network's effectiveness in mitigating near-field effects and its resilience against noise. Following this, the proposed method is applied to actual CSEM data to validate its applicability in practice. The method is subsequently tested on measured CSEM data, confirming its practical efficacy. Results from experiments indicate that, for theoretical data, the LSTM-CNN network-trained data closely aligns with simulated data, showcasing a significant improvement. Moreover, when applied to measured data, the method eradicates false high-resistance anomalies at lower frequencies. In conclusion, this deep learning-based correction method proficiently eliminates the influence of near-field effects in the CSEM, delivering practical application benefits that more accurately reflect the authentic geologic structure.

## 1. Introduction

Controlled Source Electromagnetic Method(CSEM) is a kind of artificial source electromagnetic method, because of its high efficiency, high resolution and other advantages are widely used in all kinds of underground resources, hydrology, engineering, etc. There are many kinds of controlled source electromagnetic method, such as common Controlled-source Audio-frequency Magnetotellurics (CSAMT), Time-requency Electromagnetic (TFEM) and so on. The traditional controlled-source electromagnetic method resistivity is defined in the same way as the Cagniard resistivity of the Magnetotelluric (MT), which leads to a non-plane-wave effect in the near zone and the transition zone close to the artificial transmitting source, i.e., it does not satisfy the assumption of MT plane wave [1–5]. This leads to the near zone and transition zone

in github(https://github.com/cxjcxj123/CSEM-LSTM_CNN).

**Funding:** "This work was co-supported by the China Postdoctoral Science Foundation (2022M723684), Sichuan Science and Technology Program (2022JDRC0066)," The funder provided research design, data collection, and data support in this study.

**Competing interests:** The authors have declared that no competing interests exist.

Cagniard resistivity serious distortion, mainly manifested in the double logarithmic coordinate system Cagniard resistivity curve in the low-frequency band was 45˚rise, this distortion effect is the near-field effect [6, 7]. In practical exploration, the near-field effect will largely reduce the survey depth of the Controlled Source Electromagnetic Method.

Numerous scholars have carried out in-depth studies to solve the problem of near-field effects of CSEM. Lei et al. [8] simulated the near-field effect and analyzed the laws and characteristics of its influence on CSAMT applications, concluded that the occurrence and intensity of the near-field effect depends on the location of the dipole source, and proposed three principles for determining the optimal location of a grounded dipole source in the field; H Hamdi et al. [9] calculated the far-field and near-field coefficients by calculating them from normalized frequency data in order to determine the resistance values in the near-field and transition regions; Luan et al. [10] proposed a CSAMT near-field correction method based on Newton's iterative method and genetic algorithm, which was shown to be able to suppress noise interference to better reflect geologic structural features while correcting pseudo-high-resistance anomalies due to near-field effects; Wang et al. [11] obtained more accurate targeting results by introducing an additional emission and measurement region and constraining the anomalies subject to near-field effects in the inversion; Liu et al. [12] analyzed the full-field apparent resistivity response with and without IP effects by simulating the full-field apparent resistivity effect combining electromagnetic induction and IP effects on a 1D medium. Liu et al.[13] based on the inverse function theorem, calculated and analyzed the characteristics of the full domain apparent resistivity curves of four different geologic models at different locations with respect to the variations of parameters such as stratum resistivity, and concluded that the full domain apparent resistivities computed by using the vertical magnetic induction can accurately reflect the electrical distribution law of the unused geologic models; Jia et al.[14] established a method for defining the multicomponent full-domain apparent resistivity in the aeronautical transient electromagnetic method, and proposed a corresponding full-domain apparent resistivity algorithm for each component of the magnetic field strength, realizing the computation of multicomponent and full-time apparent resistivity. There is also a direct band-source inversion without correction for near-field effects in practical applications [15–18],Or simply truncate and delete the data affected by the near-field effect [19, 20].The above methods to reduce the impact of near-field effect from different angles have achieved good results, but there are some limitations. For example, literature [11] constrains anomalies affected by near-field effects by introducing an additional emission and measurement region, but this method makes the cost of the work increase dramatically. Literature [19, 20] truncates and removes low-frequency data that is affected by near-field effects, whereas the lower the frequency, the deeper the depth of detection, so removing low-frequency data directly constrains the depth of detection. And in this paper, without increasing the cost of work and deleting the data, we avoid the complex calculation of correction coefficients by introducing deep learning to complete the correction of the data affected by the near-field effect.

At present, deep learning has made breakthrough progress in various fields, and in the field of geophysics, the application of deep learning to solve geophysical problems is also a mainstream research direction, using deep learning to solve transient electromagnetic denoising [21, 22], geodetic electromagnetic denoising [23], and aeromagnetic inversion [24], geomagnetic inversion [25–27] are even more research hotspots. Accordingly, we introduces deep learning into the near-field correction, experiments and verifies the feasibility and superiority of the method, and provides a new way of thinking for the near-field correction method, the main research results of this paper include:

Construction of large-scale and diversified near-field correction data sets. By simulating and analyzing the main influencing factors of the near-field effect and data characteristics, a

large-scale diversified geologic model with different resistivity and different number of strata is established, and the CSAMT data (with near-field effect) and MT data (without near-field effect) are simulated using the orthogonal algorithm, so as to construct the sample dataset suitable for deep-learning training.

Realization of high-precision near-field correction of simulated data based on deep learning. An end-to-end near-field correction neural network LSTM-CNN based on the combination of CNN and LSTM is constructed, which focuses on local feature extraction while enhancing the long-range information interaction capability. In the designed experiments on single-type stratigraphy, multi-type stratigraphy and noise environment simulation data, it is shown that the near-field correction results predicted by the LSTM-CNN are in high agreement with the MT data, which demonstrates that the method in this paper has a high degree of generalization as well as noise immunity.

Realization of deep learning near-field correction for real measurement data. Applying the method of this paper to the CSAMT dataset collected from a real engineering area in China, the experimental results show that after training in a large-scale diversified simulation dataset, the fitted model has a certain ubiquitous a priori ability, and after that, by migrating and learning from a small amount of measured data, the near-field correction of the rest of the measured data can be effectively realized, and the correction results do not have any obvious characteristics of near-field effect. It shows that the near-field correction based on deep learning can effectively avoid complex calculations such as correction coefficients, and can be applied to practical engineering.

## 2. Methods

### 2.1. Motivation and general framework

The near-field effect in CSEM is a common and relatively complex problem, caused by a variety of factors, such as transceiver distance, underground medium resistivity, etc., which also causes even in the same work area of the different measurement points by the degree of influence of the near-field effect and the frequency range is not the same, greatly increasing the difficulty of the correction. However, from the perspective of deep learning, the near-field effect correction can be regarded as a regression problem, the goal of the regression problem is to establish a mapping between the input and the output, where the input is the original CSEM data, which contains the influence of the near-field effect, and the output is the CSEM data that has been calibrated or does not contain near-field effect, i.e., the influence of the near-field effect is removed. Therefore, the goal of the deep learning regression model is to learn the mapping from the original CSEM data to the corrected CSEM data in order to more accurately reflect the subsurface structure while eliminating the effects of near-field effects. The method of this paper can be described as the following equation:

$$y = Net(x, m) \tag{1}$$

where x denotes the CSEM data input to the network; y denotes the corrected data predicted by the network; m denotes the network parameters in the function; and Net denotes the mapping function from x to y, which involves the design of the network architecture.

### 2.2. Data sets

We take the most representative of the Controlled Source Electromagnetic Method (CSEM), Controlled -source Audio-frequency Magnetotellurics (CSAMT), as an example for the next study. In practice, CSAMT data and MT data are basically not collected at the same time in the

same area, so the dataset here is obtained by forward simulation calculations, where different layered geologic models are first set up, and the corresponding resistivities are computed using the 1D forward algorithms for CSAMT and MT as the input data (CSAMT) as well as the labeled data (MT), respectively. In CSAMT, the electric dipole in a uniform horizontal ground electric and magnetic fields are as follows:

$$E_x = \frac{P_E \rho_1}{2\pi r^3} \left[ 3\cos^2\varphi - 2 + (1 + k_1 r)e^{-k_1 r} \right] \tag{2}$$

$$H_y = \frac{P_E}{2\pi r^2} \left[ (1 - 4\sin^2\varphi)I_1 K_1 + \frac{k_1 r}{2}\sin^2\varphi(I_0 K_1 - I_1 K_0) \right] \tag{3}$$

Where $I_0, K_0, I_1, K_1$ are the first and second class modified Bessel functions of zero and first order with zonal quantities $\frac{1}{2}k_1 r$; r is the send/receive distance; $k_1 = \sqrt{-i\omega\mu/\rho_1}$; $P_E$ is the current dipole moment $P_E = I \times AB$; $\varphi$ is the angle between the r direction and the X-axis.

In MT, the electric and magnetic fields in a homogeneous half-space are shown in the following equation, where A,B,C,D are constant of integration.

$$E_x = Ae^{ikz}e^{-i\omega t} + Be^{-ikz}e^{-i\omega t} \tag{4}$$

$$H_y = Ce^{ikz}e^{-i\omega t} + De^{-ikz}e^{-i\omega t} \tag{5}$$

Both CSAMT and MT calculate apparent resistivity according to Eq (6). Where $\mu$ is the magnetic permeability and $\omega$ is the angular frequency.

$$\rho_s = \frac{1}{\omega\mu}\left| \frac{E_x}{H_y} \right|^2 \tag{6}$$

In the forward simulation calculations, different geologic model parameters need to be set, such as the number of layers, the resistivity of each layer, and the thickness of the layer, etc. The CSAMT forward requires the setting of parameters such as the length of the transmitting source, the transceiver distance, and the frequency, while the MT forward requires only the frequency to be set. For the same geologic model, the CSAMT forward resistivity has aberrations due to the near-field effect and cannot reflect the real geologic model, while the MT forward resistivity has no aberrations as shown in Fig 1, which provides a reliable dataset production method for our study. The CSAMT 1D forward simulation uses the Dipole1D program published by Kerry Key [28, 29].Meanwhile, we utilizes Matlab platform programming

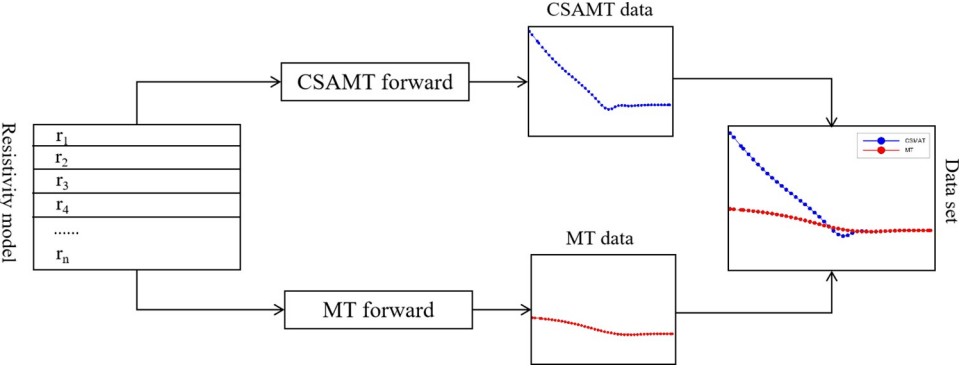

**Fig 1. Schematic diagram of data set construction.**

to realize CSAMT and MT batch normalization to meet the construction of sample dataset with large data volume.

## 2.3. Network architecture and principles

In recent years, deep learning techniques have been widely used in a range of geophysical data processing tasks, and many researchers have proposed deep learning methods based on recurrent neural networks (RNN) or convolutional neural networks (CNN) for specific tasks. Among them, LSTMs based on RNNs with improved long-range dependence problem have been applied to tasks such as geomagnetic denoising [30], transient electromagnetic denoising [31], natural earthquake magnitude prediction [32, 33], and artificial seismic first arrivals pickup [34]. Its core component, the memory cell, is able to store information and deliver it when needed, solving the gradient vanishing and gradient explosion problems of traditional RNNs when dealing with long sequences. The various gate structures of LSTM are set up to allow the memory cell to store and receive longer information, which not only alleviates the gradient vanishing problem of the RNN but also permits more efficient extraction of the features of the time-series data [34, 35].The main operational formulas of the LSTM can be described as follows:

$$
\begin{aligned}
i_t &= \sigma(W_{xi}x_t + W_{hi}h_{t-1} + W_{ci}c_{t-1} + b_i) \\
f_t &= \sigma(W_{xf}x_t + W_{hf}h_{t-1} + W_{cf}c_{t-1} + b_f) \\
c_t &= f_t c_{t-1} + i_t \tanh(W_{xc}x_t + W_{hc}h_{t-1} + b_c) \\
o_t &= \sigma(W_{xo}x_t + W_{ho}h_{t-1} + W_{co}C_t + b_o) \\
h_t &= o_t \tanh(c_t)
\end{aligned}
\tag{7}
$$

where σ denotes the Sigmoid function: $\sigma = 1/(1 + e^{-x})$; i, f, o and c denote the activation vectors of the update gate, forgetting gate, output gate and cell, respectively, and their dimensions are the same as that of the hiding vector h; tanh(ct) denotes the hyperbolic tangent function, and W denotes the weight matrix. the individual structure of the network of the LTSM is shown in Fig 2, where the meaning of each parameter is the same as that of the above equation.

The local feature learning capability and translation invariance possessed by CNN make it the architecture of choice for building deep learning networks in various fields. The CNN network commonly used for one-dimensional data is shown in Fig 3. First, multiple convolution operations are used to extract the deep abstract features of the data, and then convolution is used to gradually restore the original dimensions of the data. In this process, the sequence length of the data remains unchanged, but the network fully integrates the encoding and decoding information through the context information transfer of the jump connection. Convolutional operation is the most basic computational process in convolutional networks, and its computational process is as follows:

$$
y^l = f\left(\sum_{i=1}^{C} x_i^{l-1} \otimes w_i^l + b_i^l\right)^l
\tag{8}
$$

where f(x) denotes the activation function; yl denotes the output feature sequence of the lth layer of the network; C indicates the number of channels; $x_i^{l-1}$ denotes the input feature sequence of channel i in layer l-1;Ydenotes the convolution operation; $w_i^l$ denotes the weight value of the lth layer; $b_i^l$ denotes the bias value of the lth layer.

According to the respective advantages of LSTM and CNN, this paper combines the features of both of them and designs a deep learning network LSTM-CNN suitable for near-field

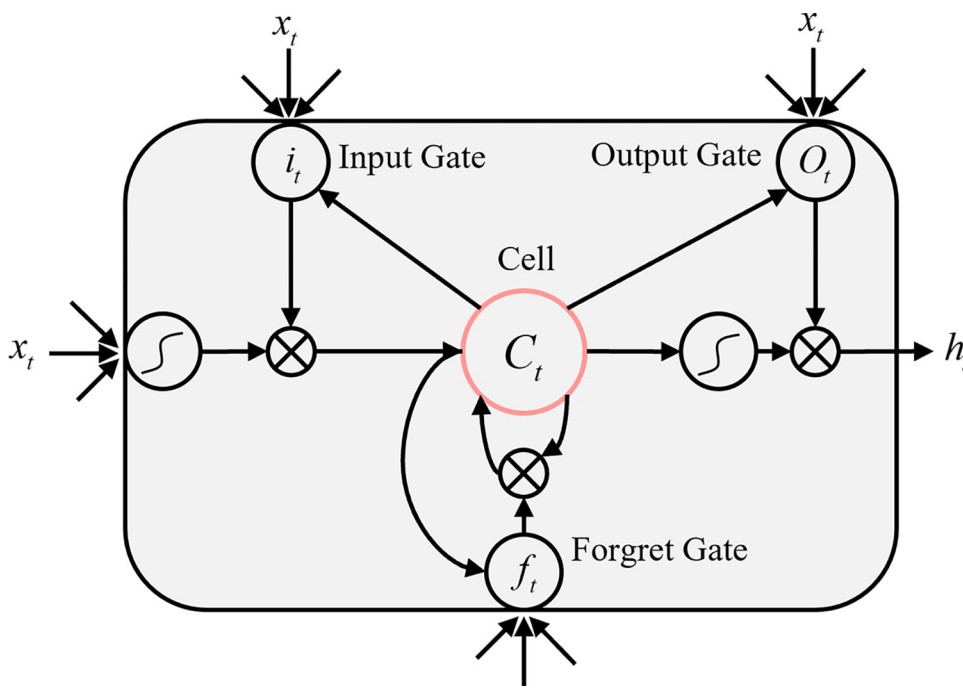

**Fig 2. LSTM cell structure diagram.**

correction, and the network architecture is shown in Fig 4. As can be seen from the figure, the LSTM-CNN network architecture consists of an encoding part with feature dimensions ranging from low to high and a decoding part with feature dimensions ranging from high to low.

The main feature extraction method of the network is convolution, and the encoding part relies on 5-stage (3, 1) convolution kernel, 1-step, (1, 0) padding convolutional layers to progressively increase the feature dimensions of the data, which enables the network to obtain feature information of the EM data in the deep dimension. The decoding part uses the corresponding convolutional layers to sequentially reduce the feature dimension of the data and gradually maps to the target dimension to realize end-to-end approach correction. In addition, since the near-field correction of EM data requires simultaneous consideration of the correlation between low-frequency and high-frequency information, we introduce LSTM to enhance the network's ability to rely on long-range information. The information interaction between long-range features is realized by concatenating the feature information computed by

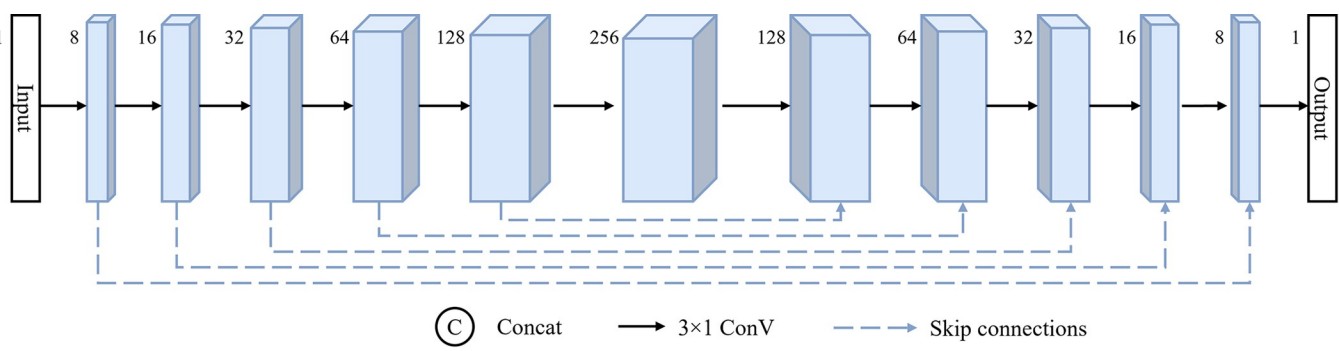

**Fig 3. Structure of the CNN network.**

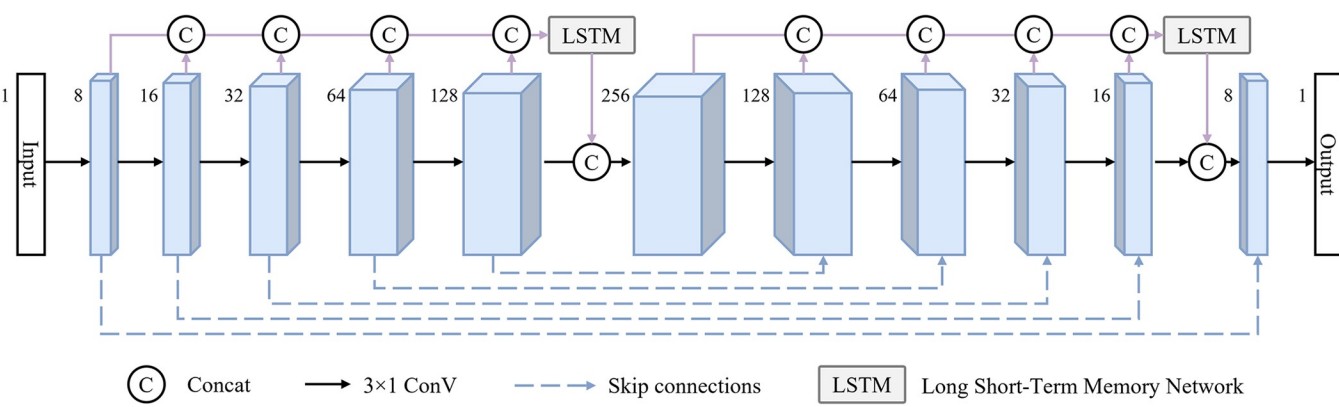

**Fig 4. Structure of the LSTM-CNN network.**

convolution at each stage in the network over the channel and inputting them together into the LSTM for global feature computation.

By combining the neural network of CNN and LSTM to take into account both local and long-range feature extraction, the LSTM-CNN proposed in this paper is able to realize the near-field correction efficiently, and it can be applied to arbitrary size data because the network does not vary the length of the data sequence. Meanwhile, in order to accelerate the fitting of the network and enhance the transmission of contextual information of the network, a jump connection with channel Concat is performed in the corresponding stage of the encoding and decoding part. The final of the network is then done by the Sigmoid function for the final output and the loss is calculated by the MSE function, the formula for the MSE loss is as follows:

$$\text{MSE Loss} = \frac{1}{N} \sum_{i=1}^{N} (x_i - y_i)^2 \tag{9}$$

where N denotes the number of samples; $x_i$ denotes the true value (MT data); and $y_i$ denotes the network predicted value.

## 2.4. Training environments, strategies, and evaluation metricse

The experiment was deployed on a PC with AMD Ryzen5 3600X processor, NVIDIA RTX 3080Ti, 16GB graphics card on a PC with Pytorch 1.8.2 + CUDA11.1 network training framework, and the specific hardware and software configurations used in the experimental environment are shown in Table 1.

In order to compare the advantages of the proposed network LSTM-CNN in this paper, we constructed both CNN and LSTM networks individually and conducted experiments in the

**Table 1. Experimental hardware and software configuration table.**

| Training hardware and software | |
|---|---|
| Programming language | Python 3.8.13 |
| Network Training Framework | Pytorch 1.8.2+ CUDA11.1 |
| CPU | AMD Ryzen5 3600X 6-core |
| GPU | RTX 3080Ti 16GB |
| Hard disk | 1TB Solid State Drive |
| Random access memory | 32G 3200 MHz |

single-type stratum, multi-type stratum, and noisy environment simulation data designed in this paper, and judged the networks by calculating the MAE values as well as the MRE values of all three in each dataset.

The trends of MRE and MAE are the same, but the range of values is different, combining the two can more comprehensively evaluate the good and bad experimental results, and analyze the advantages and disadvantages of different types of networks. MAE and MRE can be expressed as the following equation:

$$\text{MAE} = \sum_{i=0}^{N}(|y_i - x_i|) \tag{10}$$

$$\text{MRE} = \sum_{i=0}^{N}(\frac{|y_i - x_i|}{x_i}) \tag{11}$$

where xi denotes the true value; yi denotes the network predicted value; and N denotes the total number of data.The RAdam optimizer is used for each network training, and the training settings of Batch Size and initial learning rate are the same, and the experiments set a higher maximum number of training rounds (500 rounds). Adaptive learning rate scheduling strategy is used to update the learning rate to 1/2 of the current rate when the loss is no longer decreasing for 5 consecutive rounds. At the same time, this paper experiments adopt the network training method of stopping early in order to avoid the network overfitting and the network training time consuming too long.

## 3. Theoretical data experiment

### 3.1. Single geologic model dataset testing

In this section, two theoretical datasets, the three-layer geologic model as well as the ten-layer geologic model, are constructed respectively, in which the three-layer geologic model dataset establishes 4,000 models, the thickness of each layer of each model is fixed to 500m, and the resistivity of each layer of the model is randomly valued between 10 and 1,000 $\Omega$·m. The simulation frequency is $2^{-3}$~$2^{13}$Hz, with a total of 44 frequency points, and the length of the transmitting source of the CSAMT is 2km, and the transceiver distance is 5km, and 4000 sets of data samples of the three-layer geologic model CSAMT and the corresponding MT are obtained by forward simulation.

The ten-layer geologic model dataset simulates a complex geological environment, and the production method and parameters are basically the same as those of the three-layer geologic model. The difference is that the resistivity of each layer is randomly taken between 1 and 10,000 $\Omega$·m, totaling 50,000 models, and 50,000 sets of samples are obtained by forward simulation.

The established CNN, LSTM, and LSTM-CNN networks are experimented using the above two datasets, respectively. Meanwhile, in order to compare with the traditional methods, the experiments are carried out by using the Kf-Kn coefficient correction method in the traditional near-field correction method, which is introduced by the Canadian Phoenix Company [35] and is more widely used. The results are shown in Table 2. As can be seen from the table, In the three-layer geoelectric model, relative to the traditional Kf-Kn, each network shows better near-field correction results, with MRE values below 1% and much lower than the 4.66% of Kf-Kn, of which the MRE of LSTM-CNN is even as low as 0.27%, and the MAE is also excellent at 1.74, which is significantly ahead of CNN, LSTM and Kf-Kn. In the complex ten-layer geologic model, the advantage of LSTM-CNN is even more obvious, and the MAE is compared with the rest of the network with an average reduction of 10 $\Omega$·m.

**Table 2. Table of test results.**

| Geologic model | Net | MAE (Ω·m) | MRE (%) |
|---|---|---|---|
| Three-layer geologic model | CNN | 5.67 | 0.74 |
| | LSTM | 7.86 | 0.95 |
| | LSTM-CNN | 1.74 | 0.27 |
| | Kf-Kn | 33.89 | 4.66 |
| Ten-layer geologic model | CNN | 27.69 | 1.71 |
| | LSTM | 26.22 | 1.95 |
| | LSTM-CNN | 17.73 | 1.21 |
| | Kf-Kn | 443.62 | 18.53 |

Figs 5 and 6 demonstrate the comparison results of the near-field corrections of different networks, and it can be seen that the performance of each curve in the figures is consistent with the numerical results in Table 2. In the three-layer geoelectric model, each network correction results are better, only in some frequency points CNN and LSTM deviation, but the overall data features have not been lost, no matter in the near-field region or the transition region are better correction effect. Meanwhile, the traditional Kf-Kn method also has some correction effect, which is relatively good in the near-field region, but in the transition region, the method does not have any correction effect. In the ten-layer geoelectric model, LSTM-CNN predicted the correction results well, while CNN and LSTM produced large errors at some frequency points, where CNN showed local deviations, indicating that focusing on local feature extraction alone cannot be applied in complex geoelectric models. As for the traditional Kf-Kn method, the overall correction effect is relatively poor in the case of complex geologic models, and overcorrection occurs, and the corrected resistivity values are much smaller than the true values, regardless of whether they are in the near-field region or the transition region. The reason is that the method itself applies the correction method of uniform geodesy to layered geodesy, which makes its correction accuracy not high, so the correction results deviate from the real results when the distribution of strata is more complicated. It is also worth noting that, regardless of the correction method, the correction effect of the ten-layer geoelectric model is not as good as that of the three-layer geoelectric model, because the

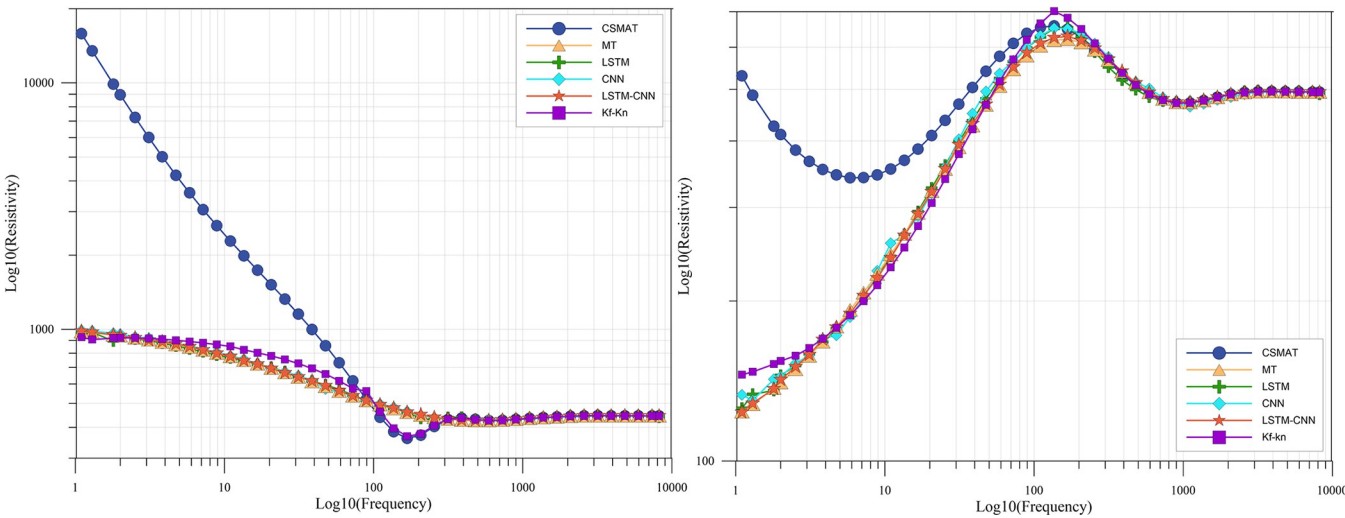

**Fig 5. Comparison of test results of three-layer geologic modeling.**

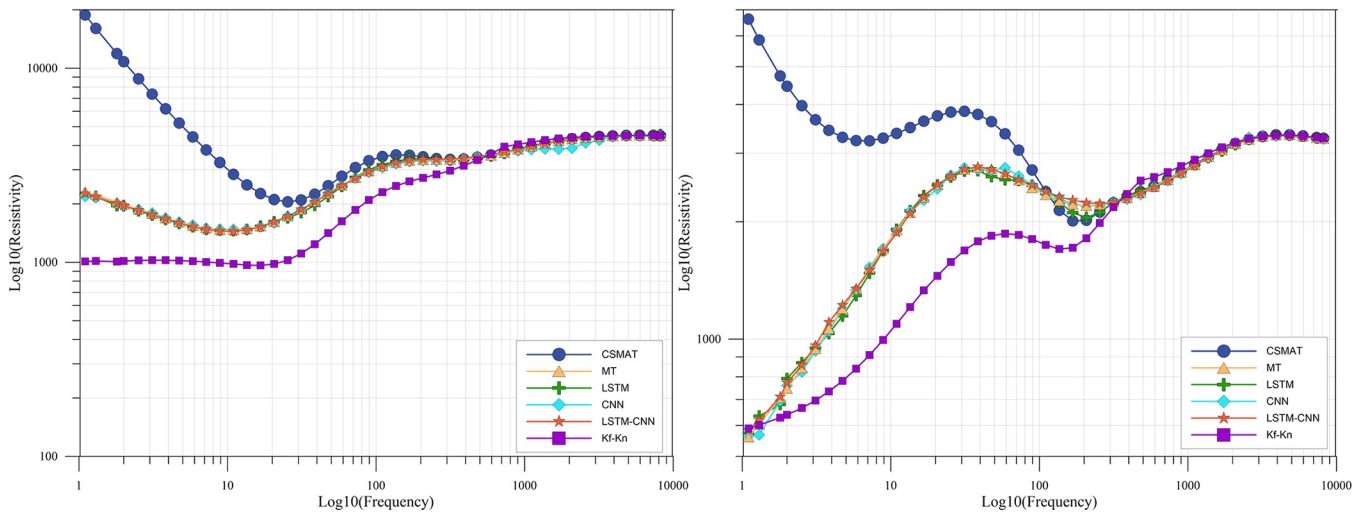

**Fig 6. Comparison of test results of ten-layer geologic modeling.**

more complex the geoelectric model is, the more complex the characteristics of the curves that are affected by the near-field effect are, and the more difficult it is to correct them.

## 3.2. Hybrid geologic model dataset testing

In practice, the resistivity curve characteristics of different measurement points in the same work area are different, and the degree of influence by the near-field effect is also inconsistent, so it is obvious that a dataset containing only one kind of geologic model data is seriously detached from reality. Therefore, in order to be more relevant to the reality, and at the same time to verify the training effect of this paper's network on multi-featured data, this section establishes several different geologic models, which are constructed into a hybrid geologic model dataset. The production method and parameters are consistent with those of the previous section, with the difference that the number of geologic model layers varies from 2 to 5, and the resistivity of each layer is randomly valued between 1 and 10,000 Ω·m. In total, 20,000 sets of sample data are obtained from the forward simulation.

Experiments were conducted on the established CNN, LSTM, LSTM-CNN networks and Kf-Kn method using the hybrid geologic model dataset, and the results are shown in Table 3. As can be seen from the table, in the 2 to 5 random layers geologic model data, the results of each network are worse than the single stratum data, in which the numerical results of CNN and LSTM decrease much more than LSTM-CNN. Due to the wide range of resistivity variations in this dataset, the MAE values are all higher than that of the single stratum data, but LSTM-CNN still achieved 1.41%, while CNN and LSTM 2.8% and 2.98%, respectively. It shows that the generalization of LSTM-CNN is better than CNN and LSTM. On the other

**Table 3. Table of test results.**

| Geologic model | Net | MAE (Ω·m) | MRE (%) |
|---|---|---|---|
| Hybrid geologic model | CNN | 93.07 | 2.80 |
| | LSTM | 84.46 | 2.98 |
| | LSTM-CNN | 41.77 | 1.41 |
| | Kf-Kn | 717.26 | 21.20 |

hand, the Kf-Kn method, shows the same overcorrection as the previous results, and the overall correction is poor.

### 3.3. Noise geologic modeling dataset testing

The above experiments demonstrate that LSTM-CNN possesses high-precision near-field correction capability for different types of simulated data. In order to verify its noise immunity, we add different levels of noise to two single geologic model datasets to simulate the actual collected low signal-to-noise ratio data, and analyzes the noise immunity of LSTM-CNN by cross-testing under different noise level data. The superposition formula by level noise is shown below:

$$noise = random[-s \times \alpha, s \times \alpha]$$
$$s' = s + noise \tag{12}$$

where s denotes the original single data; α denotes the noise level, which is taken as 5%, 10% and 15% in this paper; and s´denotes the noise-added data.

The experimental results are shown in Fig 7 and Table 4. In Fig 7, the horizontal axis represents the noise level of the training dataset, the vertical axis represents the noise level of the test

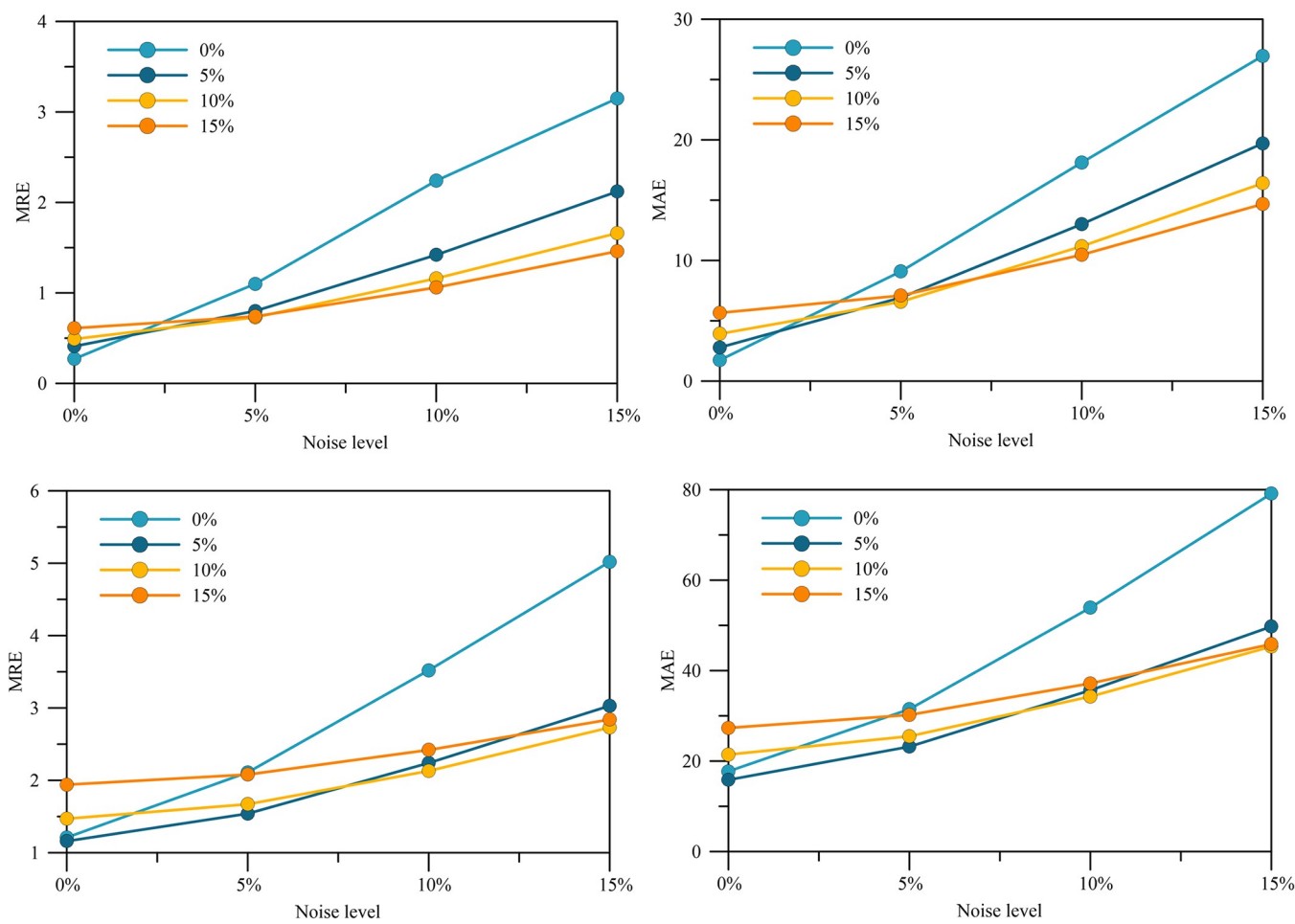

**Fig 7. Error curves for different noises.**

**Table 4. Table of noise test results.**

| | | MAE (Ω·m) | | | | MRE (%) | | | |
|---|---|---|---|---|---|---|---|---|---|
| | | 0% | 5% | 10% | 15% | 0% | 5% | 10% | 15% |
| Three-layer geologic model | 15% | 26.96 | 19.71 | 16.40 | 14.68 | 3.15 | 2.12 | 1.66 | 1.46 |
| | 10% | 18.12 | 13.01 | 11.18 | 10.47 | 2.24 | 1.42 | 1.16 | 1.06 |
| | 5% | 9.09 | 6.92 | 6.58 | 7.09 | 1.10 | 0.80 | 0.73 | 0.74 |
| | 0% | 1.74 | 2.79 | 3.92 | 5.66 | 0.27 | 0.41 | 0.49 | 0.61 |
| Ten-layer geologic model | 15% | 79.13 | 49.74 | 45.32 | 45.85 | 5.02 | 3.03 | 2.73 | 2.84 |
| | 10% | 53.91 | 35.65 | 34.27 | 37.17 | 3.52 | 2.24 | 2.13 | 2.42 |
| | 5% | 31.47 | 23.19 | 25.49 | 30.21 | 2.11 | 1.54 | 1.67 | 2.08 |
| | 0% | 17.73 | 15.85 | 21.44 | 27.33 | 1.21 | 1.16 | 1.47 | 1.94 |

dataset, and Table 4 shows the specific values corresponding to Fig 8. It should be noted that the noise levels of the training and validation sets are kept consistent during training. As can be seen from the figure, for the fitted model trained under noisy data, the higher the noise level of the test data, the faster its MRE and MAE increase. The overall curve slope is larger than the

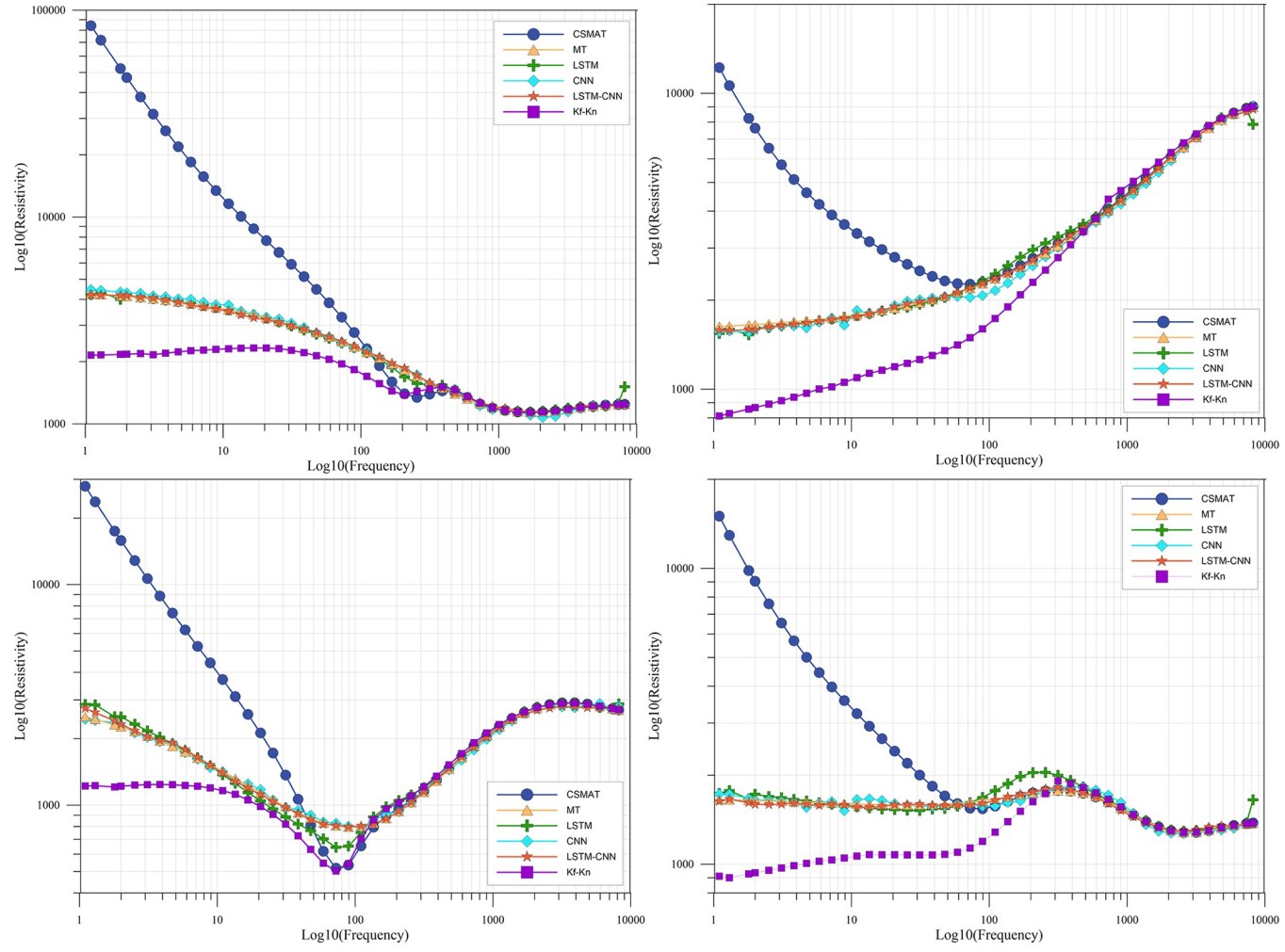

**Fig 8. Comparison of hybrid geologic model test results.**

rest of the test results trained under noisy data, and this trend is more obvious in the ten-layer geologic model. As for the fitted models trained under each noisy data, the model trained under 10% noise level showed the most stable performance among the test data with each noise level, and showed effective near-field correction results for any noise level data. Combining the test results of the two geodesic datasets, it can be concluded that the LSTM-CNN has good noise immunity, and the fitted models trained under different noise levels also show good generalization.

## 4. Application of measured data

### 4.1. Overview of the work area

The measured data is the CSAMT data of a railroad tunnel survey, the tunnel is located in Gaoligong Mountain, Yunnan Province, China, the Gaoligong Mountain range starts from Tanggula Mountain in the north of the Tibetan Plateau, enters into Yunnan from Tibet, and enters into the territory of Baoshan region after passing through the northwest of Yunnan to the Nujiang Prefecture with the length of about 215km. the main mountain range is distributed in the Nujiang River and Longchuanjiang, which is narrow in the north and wide in the south in the north-south direction, with narrow east-west and the spacing of about 15km. 15km, the altitude is more than 3000m. Schematic layout of acquisition lines and emission sources is shown in Fig 9.

The tunnel is constructed by TBM method, however, the geological conditions of the tunnel are extremely complicated, and the machine has been stuck and trapped many times during the TBM construction project, which seriously affects the construction safety and progress of the project, therefore, China Railway Academy II carried out the supplementary geological survey of GaoLiGongShan Tunnel in 2023, and carried out the related physical exploration work by using CSAMT method, and the layout of the collecting survey line and transmitting source is shown in Fig 7, in which the transmitting source 1 dipole is 1.8km in length, and the

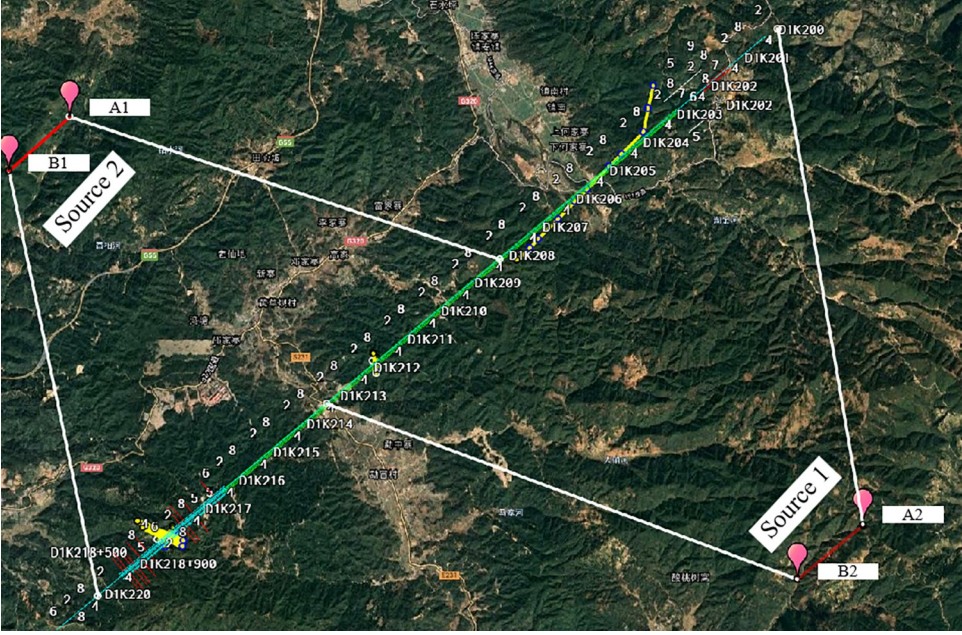

**Fig 9. Schematic layout of acquisition lines and emission sources.**

Table 5. Statistics of resistivity values of each mass in the work area.

| Lithology | Resistivity (Ω·m) |
|---|---|
| Jurassic (J) siltstone, Basalt interbedded with mudstone, siltstone and sandstone, mudstone interbedded with marl | 25~100 |
| Triassic (T) dolomitic chert, dolomite | 160~10000 |
| Devonian (D) dolomitic graywacke, dolomite, and dolomitic interbedded slate | 60~1600 |
| Silurian (S) chert sandstone and Ordovician (O) quartz sandstone and feldspathic quartz sandstone. | 25~1000 |
| Cambrian (∈) slate, kyanite interbedded with siltstone and graywacke | 40~10000 |
| Cambrian section (∈3s2) chert interbedded with marl, slate, and siltstone, and lower section (∈3s1) kyanite and slate interbedded with sandstone; | 25~1000 |
| Yanshanian granite (γ53) | 10~10000 |
| Faults or fault-influenced zones and fractured or water-rich rock masses | 1~355 |

transmitting distance is 8.9km. The length of transmitting source 1 dipole is 1.8km, and the distance between transmitting and receiving is 8.9km, and the length of transmitting source 2 dipole is 1.93km, and the distance between transmitting and receiving is 9.9km, and the transmitting sources are basically parallel to the survey line, with the maximum angle of less than 3˚. The transmitting frequency is 1.33~9594Hz, totaling 40 frequency points. The distance between measurement points is 20m, totaling 447 measurement points. The lithological geophysical characteristics of the work area are shown in Table 5.

## 4.2. Measurement dataset testing

Fig 10 shows the measured apparent resistivity curves of five measurement points. From the analysis of these measured curves, it can be seen that the overall continuity of the measured curves is better and the quality is in line with the standard, but they are more seriously affected by the near-field effect. The degree of near-field effect is also different for different measurement points, such as the 209490 measurement point enters the transition zone or near-field zone from about 20 Hz, while the 209350 measurement point is about 40 Hz. Therefore, it is obvious that the data processing and interpretation method directly adopting the natural field source can not obtain the real and reliable geologic characteristics.

There are two main problems for the measured data, one is the small amount of sample data and the other is the lack of labeled data. To address the problem of small sample data volume, we increase the sample data volume by establishing a geologic model that maximizes the conformity to the actual situation, and using the actual collection parameters to perform a large number of forward calculations. As for the problem of lack of labeled data, the strategy of this paper is to select a number of measurement points with high quality and less affected by near-field effects, and take the full domain apparent resistivity of the point as the corresponding labeled data by calculating the full domain apparent resistivity, which is too complicated to be discussed here because it is too complicated to calculate the full domain apparent resistivity. Specifically for the measured data set in this paper, firstly, according to the lithological resistivity of the work area, set the forward resistivity range of 1~10000Ω·m, the number of layers of the geologic model is set to vary randomly from 2–5 layers, and the length of the transmitter dipole, offset distance and transmitter frequency are all consistent with the actual situation, and a total of 20,000 samples of forward data are computed. At the same time, the resistivity curves of all measurement points are toured, and the measurement points less affected by the near-field effect are selected to calculate the corresponding full-domain apparent resistivity,

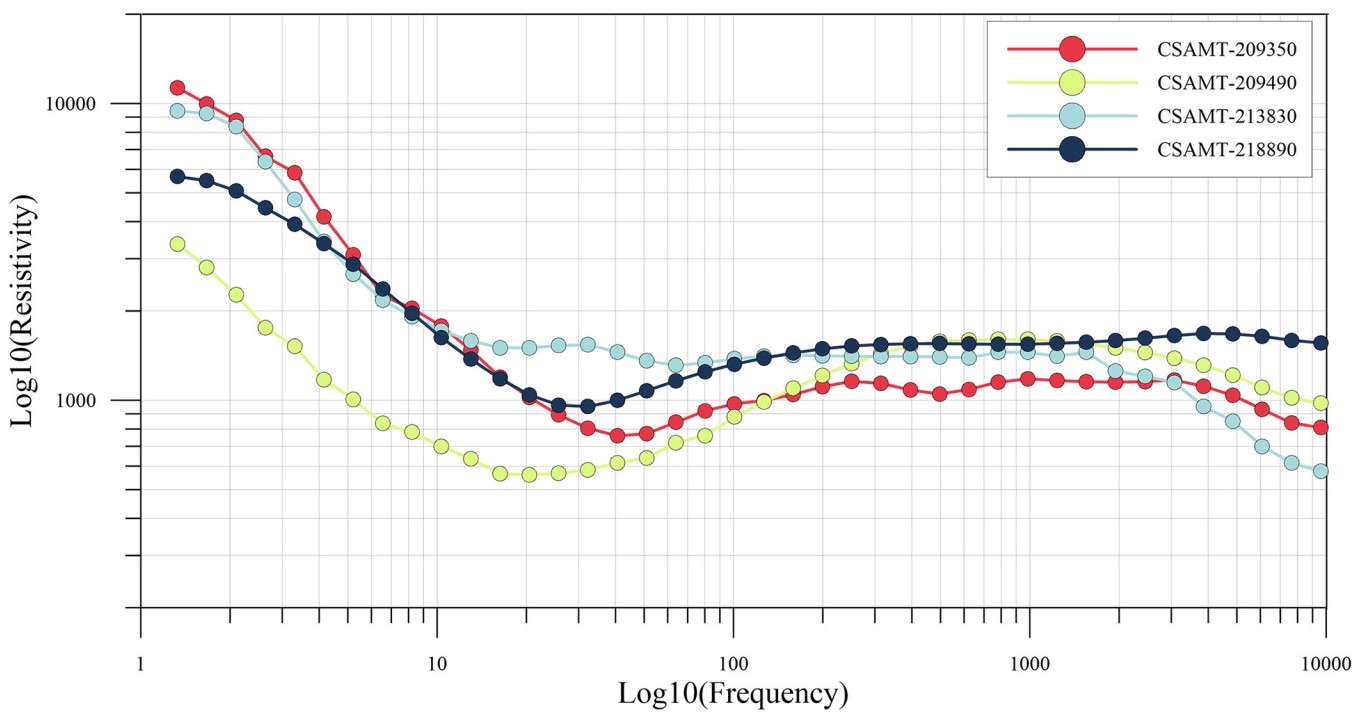

**Fig 10. Plot of apparent resistivity at selected measurement points.**

and the number of selected measurement points is 10% of the total number. Therefore, the number of sample datasets for the measurement data is 20045.

## 4.3. Analysis of results

The training experiments of the established LSTM-CNN network are conducted using the real data set, while the Kf-Kn method is used to calibrate the real data. And the correction results of some measurement points are shown in Fig 10, from which it can be seen that all the measurement points basically achieve a relatively good correction effect. For the LSTM-CNN method, in the frequency band affected by the near-field effect, the corrected apparent resistivity curves do not have obvious characteristics of the near-field effect, while in the far region, the apparent resistivity curves before and after correction remain consistent on the whole. For the Kf-Kn method, the correction effect is also very obvious, and the near-field effect is obviously improved, but there is over-correction in some measurement points, especially in the far area which is not affected by the near-field effect, the apparent resistivity is obviously lower than that before correction, such as the 209490 measurement point (Fig 11B), and even the whole frequency band appears to have the apparent resistivity much lower than that before correction after correction, such as the 218890 measurement point (Fig 11D).

The calibration effect is analyzed with the 209350 measurement point (Fig 11A) as an example. It is identified by the geology that the deep part of the 209350 measurement point is a broken water-rich rock, and its resistivity characteristic should be low resistance, while the CSAMT apparent resistivity curve of this point starts from 40Hz, and the apparent resistivity is about 700Ω·m, and then rises dramatically, and then to the lowest frequency of 1.33Hz, and the apparent resistivity is about 12000Ω·m, and the slope of the curve is greater than 1, presenting an obvious near-field effect characteristic, and the low-frequency (deep part) apparent resistivity characteristics are seriously inconsistent with the actual situation. The slope of the

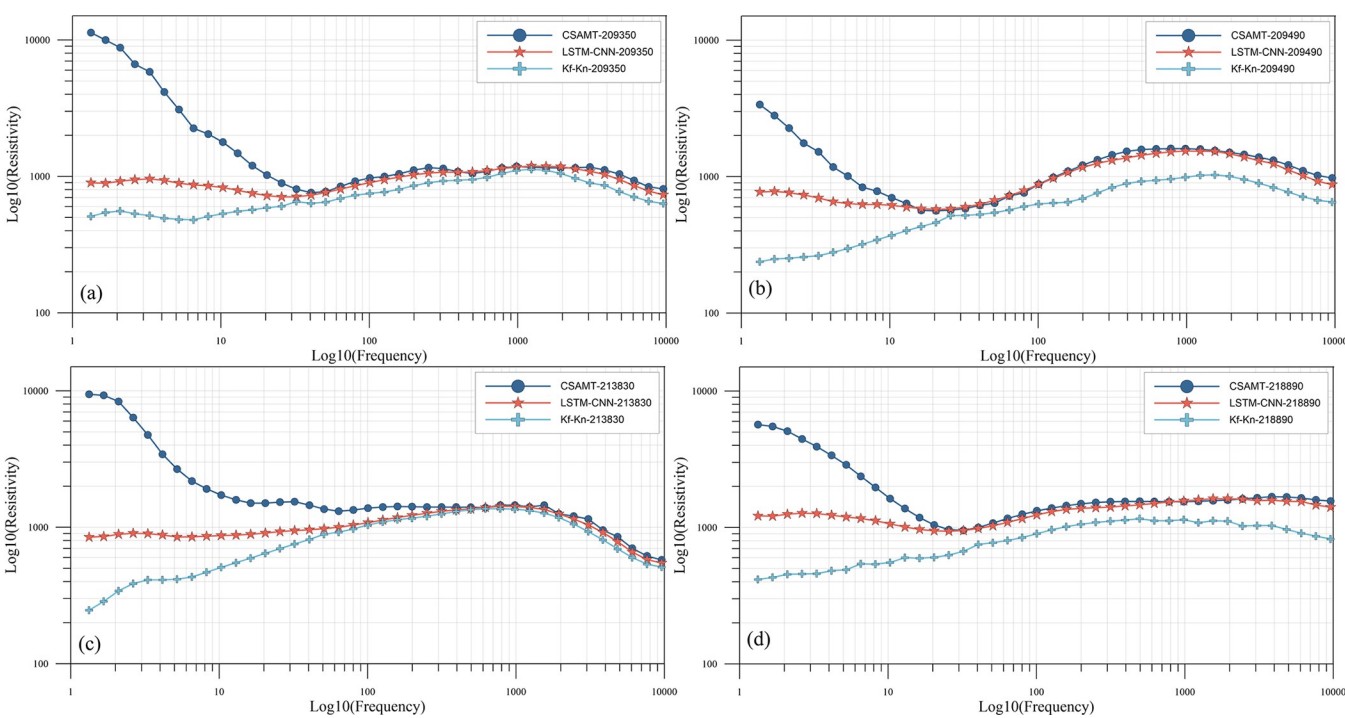

**Fig 11. Comparison of some measurement points before and after apparent resistivity correction.**

curve is greater than 1, showing obvious near-field effect characteristics, and the low-frequency (deep) apparent resistivity characteristics are seriously inconsistent with the actual situation. Corrected apparent resistivity profile using LSTM-CNN method in the near-field region from 1.33 to 40 Hz are significantly smaller than those before correction, with about 900 Ω·m at the lowest frequency of 1.33 Hz, and within the range of 700–1000 Ω·m overall, and the apparent resistivity curves are not monotonically increasing or decreasing, but show the characteristic of rising and then decreasing in the range from 1.33 to 40 Hz, which makes the information richer. It is worth noting that in the far region, the apparent resistivity curves before and after correction have some deviation in some frequency points, and this situation is common. Statistically, the deviation values of most frequency points are within the range of a few to several tens of ohm-meters, and we believe that there may be two reasons for the deviation in the far region, one is that the characteristics of the apparent resistivity curve of the measured data are different and richer, and even though forward simulation has been used to simulate the real situation as much as possible and to enrich the samples of the measured data, it is never possible to reach the real situation completely. Secondly, when the labeling data of the measured dataset are produced in the previous section, the method is to calculate the global apparent resistivity, and there is also a certain difference between the global apparent resistivity and the CSAMT apparent resistivity in the remote area, which may be one of the reasons. Nevertheless, we believe that the deviation value of some frequency points in the far region does not have a great impact on the magnitude of its apparent resistivity, and is within an acceptable range.

Fig 11 shows the comparison of the apparent resistivity sections before and after correction for all measurement points, with the x-axis being the mileage of the measurement points and the y-axis being the logarithmic frequency. The apparent resistivity cross sections before and after correction have a strong consistency on the whole, especially in the part of the middle and high frequency bands not affected by the near-field effect, the contour patterns as well as

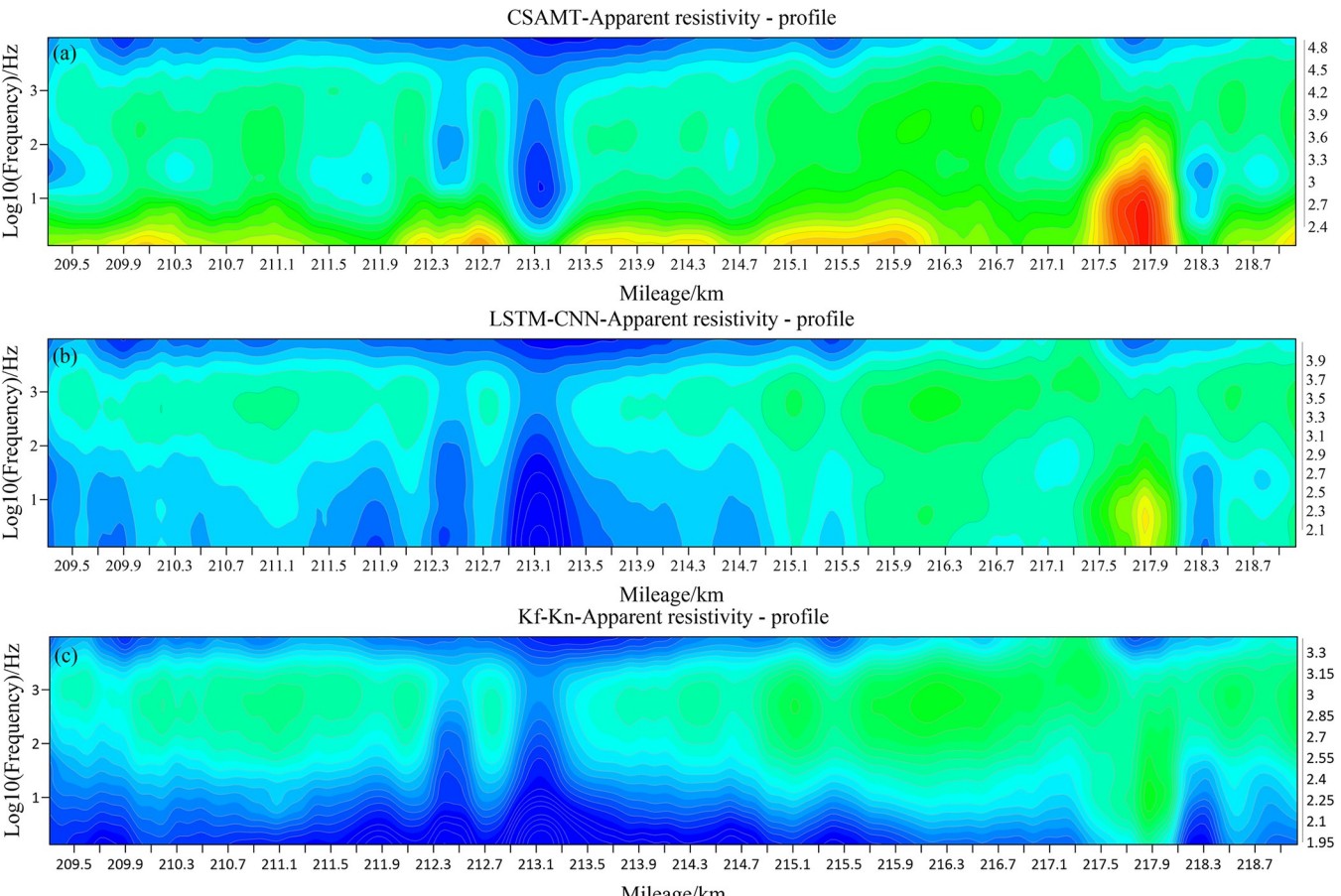

**Fig 12. Comparison of sections before and after apparent resistivity correction of measured data.** (a)CSAMT apparent resistivity profile; (b)LSTM-CNN apparent resistivity profile;(c)Kf-Kn apparent resistivity profile.

the apparent resistivity values are almost the same. However, in the CSAMT apparent resistivity section(Fig 12A), the mileage of 209700~210300m, 212100~212900m, 213300~214500m, and 214900~216100m have high resistance anomalies at the low frequency below 10 Hz, which is seriously inconsistent with the actual situation. After correction by LSTM-CNN method(Fig 12B), the false high-resistance anomalies in the above sections disappear, indicating that most of the data in the measured data set have been effectively corrected. However, it is worth noting that there are relative high resistance anomalies at mileage of 217300~218100m before and after correction, and the anomaly range is large, which indicates that the network in this paper does not simply and roughly regard the "low-frequency high-resistance" as a near-field effect, but ensures that the resistivity characteristics of each point as much as possible at the same time, and identify the near-field effect of the point and correct the anomalies. The LSTM-CNN network does not simply regard the "low frequency high resistance" as the near-field effect. In contrast, after correction by the Kf-Kn method, the apparent resistivity is lower overall, and the high-resistance anomaly at low frequencies disappears completely.

In order to further verify the calibration effect of this paper, the Winglink commercial inversion software, which is more popular at home and abroad, is used to invert the data before and after calibration, as shown in Fig 13. The inversion results before and after

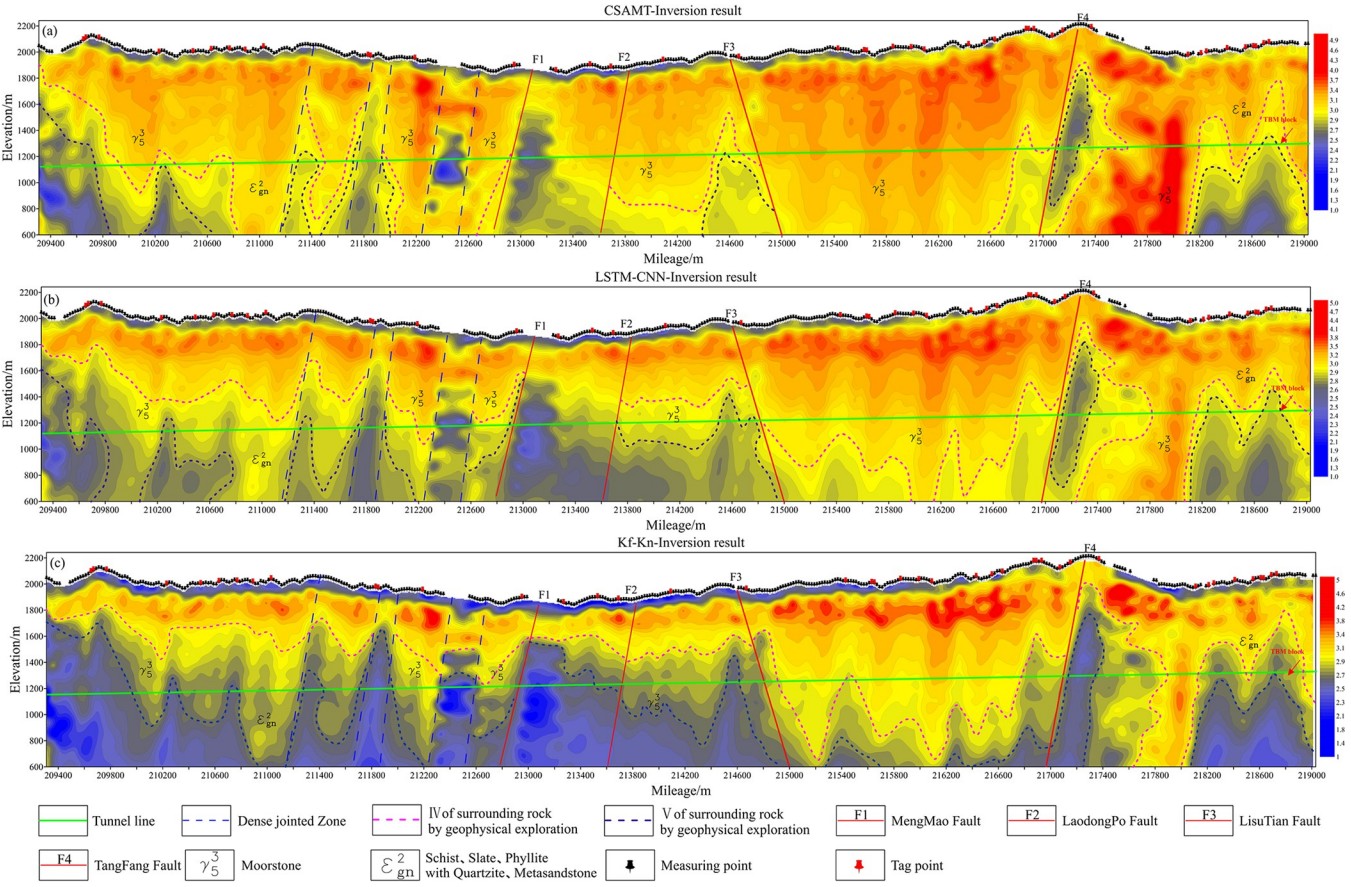

**Fig 13. Comparison of inversion results before and after correction of measured data.** (a)CSAMT inversion result; (b)LSTM-CNN inversion result; (c)Kf-Kn inversion result.

calibration are all macroscopically characterized by a steeper standing electrical distribution, with high and low resistance regions spaced apart, and overall lower resistivity values at depth. However, the LSTM-CNN corrected inversion results (Fig 13B) are richer in anomalies compared with the result before correction (Fig 13A), especially in the middle and deep parts below about 1200m above sea level, where low resistance anomalies such as fractured water-bearing bodies, joint-intensive zones, and fault-fractured zones are more obviously reflected. Influenced by the Lao Dongpo Fault and the fracture zone around the Li sutian Fault, there should be low resistance anomalies within the mileage of 213800~21500m, but the inversion result before correction (Fig 13A) is not obvious, while the LSTM-CNN corrected inversion result (Fig 13B) has obvious low resistance anomalies. At the same time, the corrected inversion results portray the anomaly range of the surrounding rock grade in a more detailed way, which corresponds more accurately with the location of the TBM jamming machine, which is of great significance for the safety of tunnel construction. However, the corrected inversion results of the Kf-Kn method, have lower resistivity values at depth and a larger range of low-resistance anomalies, but they are obviously not in line with the actual geological conditions. The reason for this is that the Kf-Kn method assumes that the subsurface is a homogeneous half-space, whereas the actual subsurface geologic conditions are complicated, leading to the over-correction of the Kf-Kn method.

## 5. Conclusions and discussion

The near-field effect significantly influences both the detection depth and efficacy of the CSEM. In recent years, deep learning technology has gained widespread application in geophysics. We integrated deep learning into near-field effect correction, achieving high-precision correction through the establishment of a large and diverse theoretical dataset. The method is further applied to real measurement data, yielding favorable results. Upon analysis and summarization of the experimental outcomes and application effects, the following conclusions are drawn:

A. The deep learning-based near-field effect correction method circumvents the intricate calculations required by traditional methods. Instead, it directly learns data characteristics through the network. Theoretical data experiments demonstrate the effective prediction of controlled source electromagnetic data subjected to near-field effects, yielding highly satisfactory correction results.

B. The LSTM-CNN network constructed in this paper adeptly combines CNN's local feature extraction and LSTM's long-range dependence. Tailored to the characteristics of controllable source electromagnetic data, this network facilitates high-precision near-field correction applicable to datasets of arbitrary size. Furthermore, it exhibits robust noise resistance and generalization capabilities.

C. Experiments applying the proposed method to real-measured controllable source electromagnetic data indicate excellent performance after learning from large-scale theoretical data enhancements. The method effectively eliminates prominent near-field effects in real-measured data, aligning the correction effect more closely with the actual geological situation.

The near-field correction method proposed in this paper introduces a fresh approach to mitigate the impact of near-field effects in the CSEM. This method employs deep learning techniques to discern and eliminate the influence of anomalous data. Comparing the outcomes with other correction methods, the approach outlined in this paper stands out for its avoidance of intricate and time-consuming calculations, simplicity in operation, and robust executability. It ensures data integrity and exhibits favorable application effects. but at present there are some limitations in the near-field correction using the deep learning method. The reason for this is that the near-field effect is a very complex problem with a large number of influencing factors and variable data distortion characteristics. On the one hand, this makes it more difficult to establish a dataset that fully meets all the characteristics of the aberrations, especially when establishing the measured dataset, the labeled data does not fully correspond to the actual situation, and there is a certain degree of bias. On the other hand, due to the complexity of the near-field effect and the characteristics of small samples have high requirements on the generalization degree and migration learning ability of the deep learning network, it is crucial to construct a deep learning network that learns through small samples and has high generalization degree and strong migration learning ability at the same time.

## Author Contributions

**Conceptualization:** Wei Luo.

**Data curation:** Xianjie Chen, Shixing Wang, Shaojun Wang.

**Formal analysis:** Xing Lan.

**Funding acquisition:** Wei Luo.

**Supervision:** Siwei Zhao.

**Validation:** Xiaokang Yin.

**Visualization:** Xianjie Chen.

**Writing – original draft:** Xianjie Chen, Peifan Jiang.

**Writing – review & editing:** Wei Luo.

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
