## [Decision Letter · Decision Letter 0]

18 Mar 2024

PONE-D-24-01179Deep learning-based near-field effect correction method for controlled source electromagnetic method and applicationPLOS ONE

Dear Dr. Chen,

Thank you for submitting your manuscript to PLOS ONE. After careful consideration, we feel that it has merit but does not fully meet PLOS ONE’s publication criteria as it currently stands. Therefore, we invite you to submit a revised version of the manuscript that addresses the points raised during the review process.

We look forward to receiving your revised manuscript.

Kind regards,

Praveen Kumar Donta, Ph.D.

Academic Editor

PLOS ONE

Journal Requirements:

4. Thank you for stating the following financial disclosure: "This work was co-supported by the China Postdoctoral Science Foundation 

(2022M723684), Sichuan Science and Technology Program (2022JDRC0066),"

5. We note that you have indicated that there are restrictions to data sharing for this study. PLOS only allows data to be available upon request if there are legal or ethical restrictions on sharing data publicly. For more information on unacceptable data access restrictions, please see http://journals.plos.org/plosone/s/data-availability#loc-unacceptable-data-access-restrictions. 

6. We note that Figure 8 in your submission contain [map/satellite] images which may be copyrighted. All PLOS content is published under the Creative Commons Attribution License (CC BY 4.0), which means that the manuscript, images, and Supporting Information files will be freely available online, and any third party is permitted to access, download, copy, distribute, and use these materials in any way, even commercially, with proper attribution. For these reasons, we cannot publish previously copyrighted maps or satellite images created using proprietary data, such as Google software (Google Maps, Street View, and Earth). For more information, see our copyright guidelines: http://journals.plos.org/plosone/s/licenses-and-copyright.

a. You may seek permission from the original copyright holder of Figure 8 to publish the content specifically under the CC BY 4.0 license.  

Reviewers' comments:

Reviewer's Responses to Questions

**Comments to the Author**

1. Is the manuscript technically sound, and do the data support the conclusions?

Reviewer #1: Partly

Reviewer #2: Partly

Reviewer #3: Partly

2. Has the statistical analysis been performed appropriately and rigorously? 

Reviewer #1: No

Reviewer #2: Yes

Reviewer #3: No

3. Have the authors made all data underlying the findings in their manuscript fully available?

Reviewer #1: No

Reviewer #2: Yes

Reviewer #3: Yes

4. Is the manuscript presented in an intelligible fashion and written in standard English?

Reviewer #1: No

Reviewer #2: Yes

Reviewer #3: Yes

5. Review Comments to the Author

Reviewer #1: Before presenting the LSTM CNN network topology in Figure 3, the traditional CNN should be presented first and analysed for readers to have insight of the background of the work.

Where is the confusion matrix, trained, untrained and best fit data for the study.

The metric of evaluation based on accuracy, Fscore, sensitivity, and selectivity are missing.

How are these results better than those by AlexNet, GoogleNet and VGG16? No detailed comparisons were carried out.

Reviewer #2: Based on the available content from the documents, here are some suggestions for improvements for the manuscript titled "Deep learning-based near-field effect correction method for controlled source electromagnetic method and application":

Clarity and Language: Ensure that the manuscript is written in clear and unambiguous professional English throughout. Review the manuscript for any grammatical or language errors and make necessary corrections.

Literature References: Check if the manuscript provides sufficient field background/context by including relevant literature references. Ensure that the references are up-to-date and cover the relevant research in the field.

Article Structure: Evaluate the professional article structure of the manuscript. Ensure that it follows a logical flow, with clear sections such as Introduction, Methods, Results, and Discussion. Review the manuscript for coherence and organization.

Figures and Tables: Assess the figures and tables included in the manuscript. Ensure that they are clear, well-labeled, and effectively support the findings and arguments presented in the text. Consider adding any additional figures or tables that may enhance the understanding of the research.

Data Availability: Verify if the manuscript includes a clear Data Availability Statement. Ensure that the statement describes where the data can be found and provides specific details such as URLs, accession numbers, or DOIs. If the data are not publicly available, provide appropriate explanations for restricted access.

Results and Definitions: Review the formal results section of the manuscript. Ensure that all terms and theorems are clearly defined. Additionally, provide detailed proofs for any claims or assertions made in the manuscript to enhance the rigor and clarity of the research.

Reviewer #3: I regret to inform that I recommend rejection of this manuscript for publication. While the topic is of significant interest and the attempt to apply deep learning to enhance near-field effect correction in controlled source electromagnetic methods is commendable, there are several critical issues that lead to this decision.

Firstly, the manuscript lacks a comprehensive comparison with existing methodologies beyond deep learning approaches. While the introduction and literature review highlight the significance of addressing near-field effects and mention some existing solutions, the manuscript does not provide a detailed comparison of the proposed LSTM-CNN model's performance against these conventional methods. Such a comparison is crucial to establish the novelty and superiority of the proposed solution.

Secondly, the validation of the proposed method, particularly on real-world data, is insufficiently rigorous. The manuscript presents results from the application of the method to both simulated and actual measurement data. However, the methodology for selecting and preparing these datasets, especially the real-world dataset, is not described in sufficient detail to ensure reproducibility. Moreover, the discussion on the limitations of the dataset and the potential impact of these limitations on the findings is lacking.

Thirdly, there are concerns regarding the data availability and the reproducibility of the results. The manuscript states that some data cannot be shared publicly due to company regulations, which restricts the ability of the research community to fully evaluate and replicate the study's findings. This limitation is significant, especially for a journal like PLOS ONE, which values openness and transparency in scientific research.

Lastly, the manuscript would benefit significantly from a more critical discussion of the results, including limitations of the deep learning approach, potential biases in the dataset, and how these might affect the generalizability of the findings. The conclusion section presents a very positive view of the proposed method's capabilities without sufficiently addressing these critical aspects.

In summary, while the manuscript tackles an important problem and proposes an innovative solution, the aforementioned concerns regarding methodology comparison, data validation, availability, and discussion of limitations necessitate substantial revisions. It is recommended that the authors address these issues comprehensively before considering resubmission to a peer-reviewed journal.

6. PLOS authors have the option to publish the peer review history of their article (what does this mean?). If published, this will include your full peer review and any attached files.

Reviewer #1: No

Reviewer #2: No

Reviewer #3: **Yes: **Batyrkhan Omarov

---

## [Author Response · Author response to Decision Letter 0]

17 May 2024

Dear Managing Editor:

Thank you for giving me the opportunity to submit a revised draft of my manuscript titled Deep learning-based near-field effect correction method for Controlled Source Electromagnetic Method and application to Plos One. We appreciate the time and effort that you and the reviewers have dedicated to providing your valuable feedback on my manuscript. We are grateful to the reviewers for their insightful comments on my paper. We have been able to incorporate changes to reflect most of the suggestions provided by the reviewers. We have highlighted the changes within the manuscript.

Here is a point-by-point response to the reviewers’ comments and concerns.

Reviewer #1: 1.Before presenting the LSTM CNN network topology in Figure 3, the traditional CNN should be presented first and analysed for readers to have insight of the background of the work.

2.Where is the confusion matrix, trained, untrained and best fit data for the study.

3.The metric of evaluation based on accuracy, Fscore, sensitivity, and selectivity are missing.

4.How are these results better than those by AlexNet, GoogleNet and VGG16? No detailed comparisons were carried out.

Response:1. Thank you for pointing this out. We agree with this comment. Our proposed LSTM-CNN network architecture mainly introduces the LSTM module on the basis of the classical CNN architecture, and fuses the multi-scale coded and decoded information through LSTM to enhance the feature extraction capability of the network.Therefore, we did not show the traditional CNN architecture when we put together the manuscript, and your reminder is right, we now add the network topology diagram of the classical CNN architecture used in this paper, shown in Fig. 1, and analyze it for the reader's benefit.

Fig1 Structure of the CNN network

2.For the confusion matrix, we regard the deep learning-based controlled source electromagnetic near-field effect correction method as a regression task, and we normalize the data at the input and output, so when calculating the performance indexes, we can not derive the relevant indexes such as FP, TP, etc. Through the research of related papers and comprehensive consideration, we believe that the use of MAE and MRE as quantitative indexes to measure the method performance is the most appropriate.In this paper, the network almost does not involve additional hyper-parameters that need to be set by human beings, the only parameters that need to be set by human beings are (Initial Learning Rate, Batch Size), and the optimizer also adopts the most common RAdam.

3.Similar to question 2, through the research of related papers and comprehensive consideration, we believe that it is most appropriate to use MAE and MRE as quantitative indicators to measure the performance of the method. At the same time, applying them to real data is our ultimate goal, so we show the effect of real data processing, and it is a common practice in the industry to show the effectiveness of the method from the perspective of visualization, and we hope that you can recognize and accept our approach.

4.AlexNet, GoogleNet and VGG16 are mostly very early classical networks, which were initially CNN-like networks for image classification and cannot be directly used for this task. In this paper, we adopt a more general comparison of network performance, i.e., CNN network, LSTM network and CNN combined with LSTM network, and we discuss the advantages and disadvantages of these three networks from a broader perspective through experiments, and finally conclude the effectiveness of CNN-LSTM, so we do not compare the networks proposed in this paper with AlexNet, GoogleNet and VGG16 for comparison.

Reviewer #2: Based on the available content from the documents, here are some suggestions for improvements for the manuscript titled "Deep learning-based near-field effect correction method for controlled source electromagnetic method and application":

1.Clarity and Language: Ensure that the manuscript is written in clear and unambiguous professional English throughout. Review the manuscript for any grammatical or language errors and make necessary corrections.

2.Literature References: Check if the manuscript provides sufficient field background/context by including relevant literature references. Ensure that the references are up-to-date and cover the relevant research in the field.

3.Article Structure: Evaluate the professional article structure of the manuscript. Ensure that it follows a logical flow, with clear sections such as Introduction, Methods, Results, and Discussion. Review the manuscript for coherence and organization.

4.Figures and Tables: Assess the figures and tables included in the manuscript. Ensure that they are clear, well-labeled, and effectively support the findings and arguments presented in the text. Consider adding any additional figures or tables that may enhance the understanding of the research.

5.Data Availability: Verify if the manuscript includes a clear Data Availability Statement. Ensure that the statement describes where the data can be found and provides specific details such as URLs, accession numbers, or DOIs. If the data are not publicly available, provide appropriate explanations for restricted access.

6.Results and Definitions: Review the formal results section of the manuscript. Ensure that all terms and theorems are clearly defined. Additionally, provide detailed proofs for any claims or assertions made in the manuscript to enhance the rigor and clarity of the research.

Response:1. We agree with this and have incorporated your suggestion throughout the manuscript. We have thoroughly reviewed the entire manuscript to ensure the professionalism of the writing and language, and have identified and corrected a number of grammatical and linguistic errors.

2.We have checked and ensured that sufficient background in the field has been provided, with the references on near-field correction containing the classic literature on related methods as well as the most recent literature in the last few years, and on deep learning the most recent literature in the last two years

3.For the article structure, we refer to the article structure of the published papers in journals, which mainly includes introduction, methodology, theoretical data experiment, application of measured data, and conclusion and discussion. Among them, the introduction mainly introduces the background of the field and the main problems, the method mainly introduces the measures we have taken to address the above problems, the theoretical data experiment is to verify the feasibility and superiority of our proposed method through the data simulated in the forward simulation, the application of the actual data is to verify the practicability of the method, and the conclusions and discussions are to summarize the work we have done and the results of our work.

4.We checked the charts and graphs in the article to make sure that all the images had a high degree of clarity, and we can follow up by providing all the images in vector format.

5.To address the issue of data sharing, the data in our paper are publicly available, including the theoretical dataset as well as the measured dataset. If necessary, we can also disclose our related program code.

6.We re-examined the results section carefully and checked the terminology included throughout the text to ensure that all terms were clearly defined to ensure that they were professional and accurate.

Reviewer #3: I regret to inform that I recommend rejection of this manuscript for publication. While the topic is of significant interest and the attempt to apply deep learning to enhance near-field effect correction in controlled source electromagnetic methods is commendable, there are several critical issues that lead to this decision.

Firstly, the manuscript lacks a comprehensive comparison with existing methodologies beyond deep learning approaches. While the introduction and literature review highlight the significance of addressing near-field effects and mention some existing solutions, the manuscript does not provide a detailed comparison of the proposed LSTM-CNN model's performance against these conventional methods. Such a comparison is crucial to establish the novelty and superiority of the proposed solution.

Secondly, the validation of the proposed method, particularly on real-world data, is insufficiently rigorous. The manuscript presents results from the application of the method to both simulated and actual measurement data. However, the methodology for selecting and preparing these datasets, especially the real-world dataset, is not described in sufficient detail to ensure reproducibility. Moreover, the discussion on the limitations of the dataset and the potential impact of these limitations on the findings is lacking.

Thirdly, there are concerns regarding the data availability and the reproducibility of the results. The manuscript states that some data cannot be shared publicly due to company regulations, which restricts the ability of the research community to fully evaluate and replicate the study's findings. This limitation is significant, especially for a journal like PLOS ONE, which values openness and transparency in scientific research.

Lastly, the manuscript would benefit significantly from a more critical discussion of the results, including limitations of the deep learning approach, potential biases in the dataset, and how these might affect the generalizability of the findings. The conclusion section presents a very positive view of the proposed method's capabilities without sufficiently addressing these critical aspects.

In summary, while the manuscript tackles an important problem and proposes an innovative solution, the aforementioned concerns regarding methodology comparison, data validation, availability, and discussion of limitations necessitate substantial revisions. It is recommended that the authors address these issues comprehensively before considering resubmission to a peer-reviewed journal.

Response:We feel great thanks for your professional review work on our article. As you are concerned, there are several problems that need to be addressed. According to your nice suggestions, we have made extensive corrections to our previous draft, the detailed corrections are listed below.

Firstly. you are right that initially we did not consider the comparison of our proposed LSTM-CNN with traditional methods, which was poorly thought out by us.In response to your comments, we have extensively added comprehensive comparisons with conventional methods in Section 3, "Theoretical Data Experiment", and in Section 4, "Application of Measured Data".Of course, in order to increase the comparability, we choose the Kf-Kn coefficient correction method, which is the most widely used and representative of the traditional methods.

Fig2 is a comparison plot of the correction effect of the three-layer model in the theoretical data, and it can be seen that the deep learning methods (CNN, LSTM, and LSTM-CNN) are all in high agreement with the results of MT, whereas the results of Kn-Kf are basically uncorrected in the transition region, and there is a certain correction in the far-field region, but it also does not reach the desirable correction effect in some frequency points.

Fig3 is a comparison plot of the correction effect of the ten-layer model in the theoretical data. In the more complex ten-layer model, the deep learning method still has a satisfactory correction effect, but the Kf-Kn method shows over-correction results in both the transition region and the far-field region, that is, the resistivity value is corrected too low.

Then, in order to compare the correction effect in the measured data, we also in the original basis, using Kf-Kn on is the measured data for correction, as shown in Figure 4, you can see in the measured data Kn-kf correction results, the same to the apparent resistivity correction is very low, and even the whole frequency band are relatively low.

Fig2. Comparison of test results of three-layer geologic modeling

Fig3. Comparison of test results of ten-layer geologic modeling

Fig4 Comparison of some measurement points before and after apparent resistivity correction

Through literature research and analysis of results, We believe that the reason for these results is that the traditional theoretical premise of Kf-Kn assumes that the subsurface medium is homogeneous.Therefore, in the simple three-layer model calibration experiments, Kf-Kn has a certain calibration effect, but in the more complex ten-layer model as well as the measured data calibration experiments, the calibration effect of the Kf-Kn method is relatively unsatisfactory, and the phenomenon of over-correction occurs.In order to more rigorously compare the correction effect of LSTM-CNN with the Kf-Kn method, we used the Kf-Kn-corrected data with inversion parameters and methods consistent with those of LSTM-CNN.By comparing the inversion results, we believe that it is still the LSTM-CNN-corrected data inversion results are more in line with the actual geological situation, please refer to the latest submitted manuscript for the detailed analysis.

Then, through a comprehensive comparison experiment between LSTM-CNN and traditional Kf-Kn, we believe that LSTM-CNN is far superior to traditional Kf-kn in terms of correction effect.

Secondly. the issues you raised about our insufficiently descriptive preparation and selection of datasets have also been targeted for revision.For the preparation of the theoretical dataset, we supplement the one-dimensional orthonormalization formulas and methods for CSAMT (input data) as well as MT (label data) in the dataset section of Section 2.2 of the manuscript, as well as stating that the CSAMT orthonormalization program that we use is the one publicly available from Kerry Key.Also when constructing the actual data set, since it is not realistically possible to collect CSAMT and MT data at the same location at the same time.Therefore, the method we use is to first build a geoelectric model that is as realistic as possible, based on the resistivity statistics of the actual work area.Then the acquisition parameters of the real data are used for the ortho calculation to obtain a number of ortho data, and finally 10% of the real data and its full domain apparent resistivity are added to the ortho data as the training set of the real data, while the rest of the real data are used as the test set.Although this approach is still at variance with the actual situation, it is the best way we can find at present.

Thirdly, for the data sharing issue, after communicating with the company, considering the openness and transparency of the plos one journal, we can share and disclose all the data involved in the article, including the theoretical dataset as well as the measured dataset, and at the same time, we are also willing to share the program and code involved in the paper.

Lastly, our current work is a preliminary exploration of solving the near-field correction problem using deep learning methods, and we have taken some results that we think are more promising, which can provide another way for solving the near-field correction problem. Of course, at present, there are some limitations in the near-field correction using deep learning methods. The reason is that the near-field effect is a very complex problem, which has more influencing factors and variable data distortion characteristics. On the one hand, it causes more difficulties in establishing data sets, and it is difficult to establish data sets that fully comply with all the aberration characteristics, especially when establishing the real measurement data set, the labeled data does not fully comply with the actual situation, and there is a certain degree of bias. On the other hand, due to the complexity of the near-field effect and the characteristics of small samples have high requirements on the generalization degree and migration learning ability of the deep learning network, so it is crucial to construct a deep learning network that learns through small samples and has high generalization degree and strong migration learning ability at the same time.

In summary,

---

## [Decision Letter · Decision Letter 1]

30 May 2024

PONE-D-24-01179R1Deep learning-based near-field effect correction method for controlled source electromagnetic method and applicationPLOS ONE

Dear Dr. Chen,

Thank you for submitting your manuscript to PLOS ONE. After careful consideration, we feel that it has merit but does not fully meet PLOS ONE’s publication criteria as it currently stands. Therefore, we invite you to submit a revised version of the manuscript that addresses the points raised during the review process.

We look forward to receiving your revised manuscript.

Kind regards,

Praveen Kumar Donta, Ph.D.

Academic Editor

PLOS ONE

Journal Requirements:

Additional Editor Comments:

The authors are requested to address the following comments before recommending it for publication.

1. The literature of this paper is weak. It is recommended to consider most recent studies and discuss their limitations. It is also requested to provide which of these limitations are addressed in this paper.

2. The discussions related to the plots, such as possible reasons noticed during experiments to gain superior performance of the proposed work over others are listed. Primary benefits, challenges, limitations are discussed.

3. Some of the mathematical formulae are available in published papers. For example, Eq. (7 and 8) available in published works maybe in Land subsidence prediction using recurrent neural networks. It is recommended to acknowledge the formulae used from the published works appropriately.

4. Provide the source code link (footnote or as a weblink citation) within the paper through GITHUB for a transparent and visibility for the implementations.

Reviewers' comments:

Reviewer's Responses to Questions

**Comments to the Author**

1. If the authors have adequately addressed your comments raised in a previous round of review and you feel that this manuscript is now acceptable for publication, you may indicate that here to bypass the “Comments to the Author” section, enter your conflict of interest statement in the “Confidential to Editor” section, and submit your "Accept" recommendation.

Reviewer #1: All comments have been addressed

Reviewer #2: All comments have been addressed

2. Is the manuscript technically sound, and do the data support the conclusions?

Reviewer #1: Yes

Reviewer #2: Yes

3. Has the statistical analysis been performed appropriately and rigorously? 

Reviewer #1: Yes

Reviewer #2: Yes

4. Have the authors made all data underlying the findings in their manuscript fully available?

Reviewer #1: Yes

Reviewer #2: No

5. Is the manuscript presented in an intelligible fashion and written in standard English?

Reviewer #1: Yes

Reviewer #2: Yes

6. Review Comments to the Author

Reviewer #1: No comments. The authors have addressed the comments to some extent.

Reviewer #2: Previous comments have been addressed satisfactory. So, in my personal opinion, the paper can be accepted.

7. PLOS authors have the option to publish the peer review history of their article (what does this mean?). If published, this will include your full peer review and any attached files.

Reviewer #1: No

Reviewer #2: No

---

## [Author Response · Author response to Decision Letter 1]

30 Jun 2024

Dear Dr. Donta:

Thank you for the academic editor and reviewer's comments concerning our manu entitled "Deep learning-based near-field effect correction method for Controlled Source Electromagnetic Method and application". Those comments are all valuable and very helpful for revising and improving our paper, as well as the important guiding significance to our researches. We have studied comments carefully and have made correction which we hope meet with approval. Revised portion are marked in blue and yellow in the paper. The main corrections in the paper and the responds to the reviewer's comments are as flowing:

1. The literature of this paper is weak. It is recommended to consider most recent studies and discuss their limitations. It is also requested to provide which of these limitations are addressed in this paper.

Response:Thank you for your comments. We supplemented the introduction section with a description of the existing controllable-source EM method of near-field correction and discussed its limitations, such as increasing the cost of the work and decreasing the effective detection depth. The method in this paper, however, can effectively avoid the above problems and does not require additional emission sources or deletion of low-frequency data.

2.The discussions related to the plots, such as possible reasons noticed during experiments to gain superior performance of the proposed work over others are listed. Primary benefits, challenges, limitations are discussed.

Response：Thank you for your comments. You are right that we have discussed the plots as well as the experiments extensively in the paper, and have made further modifications and additions to the original. For example, in subsection 3.1, we have added a discussion on the fact that the more complex the model is, the more difficult it is to calibrate it. At the same time, we have fully revised 5.Conclusion and Discussion sections to make the language more condensed and streamlined.

3.Some of the mathematical formulae are available in published papers. For example, Eq. (7 and 8) available in published works maybe in Land subsidence prediction using recurrent neural networks. It is recommended to acknowledge the formulae used from the published works appropriately.

Response：Thanks for your comments, we checked and cited the relevant papers based on the information you provided. Also carefully checked the other formulas to make sure the same problem does not occur .

4.Provide the source code link (footnote or as a weblink citation) within the paper through GITHUB for a transparent and visibility for the implementations.

Response：Thanks to your comments, we have uploaded the source code of our build network as well as the empirical data and other related files to GITHUB and inserted a link to it in the conclusion section of the paper (https://github.com/cxjcxj123/CSEM-LSTM_CNN). In addition, the data covered in the paper is also made public in Google Cloud Drive(https://drive.google.com/drive/folders/1d8Yfd2jvDWV0nnzfgduw8ezxflEEwdRf?usp=drive_link).

---

## [Decision Letter · Decision Letter 2]

12 Jul 2024

PONE-D-24-01179R2Deep learning-based near-field effect correction method for controlled source electromagnetic method and applicationPLOS ONE

Dear Dr. Chen,

Thank you for submitting your manuscript to PLOS ONE. After careful consideration, we feel that it has merit but does not fully meet PLOS ONE’s publication criteria as it currently stands. Therefore, we invite you to submit a revised version of the manuscript that addresses the points raised during the review process.

We look forward to receiving your revised manuscript.

Kind regards,

Jinran Wu, PhD

Academic Editor

PLOS ONE

Journal Requirements:

Additional Editor Comments:

Please make changes with advice from Reviewer 4.

Reviewers' comments:

Reviewer's Responses to Questions

**Comments to the Author**

1. If the authors have adequately addressed your comments raised in a previous round of review and you feel that this manuscript is now acceptable for publication, you may indicate that here to bypass the “Comments to the Author” section, enter your conflict of interest statement in the “Confidential to Editor” section, and submit your "Accept" recommendation.

Reviewer #1: All comments have been addressed

Reviewer #4: (No Response)

2. Is the manuscript technically sound, and do the data support the conclusions?

Reviewer #1: Yes

Reviewer #4: Yes

3. Has the statistical analysis been performed appropriately and rigorously? 

Reviewer #1: Yes

Reviewer #4: Yes

4. Have the authors made all data underlying the findings in their manuscript fully available?

Reviewer #1: Yes

Reviewer #4: Yes

5. Is the manuscript presented in an intelligible fashion and written in standard English?

Reviewer #1: Yes

Reviewer #4: No

6. Review Comments to the Author

Reviewer #1: No comments

Reviewer #4: Dear authors,

Couple of question:

1. What kind of inversion did you do for the original CSAMT data? MT or Controlled source inversion including the transition zone effect? It the MT only, it is not a verification. If you use Dipole1D, please do full controlled source inversion to verify your results after NN transformation.

2. Why do you do the correction of the near (transition zone) effect and not the true CSAMT inversion (even 1D).

3. Why do you analyze the Kf-Kn results if all your early tests shown that this net give bad results?

---

## [Author Response · Author response to Decision Letter 2]

28 Jul 2024

Dear Dr. Wu:

Thank you for the academic editor and reviewer's comments concerning our manu entitled "Deep learning-based near-field effect correction method for Controlled Source Electromagnetic Method and application". Those comments are all valuable and very helpful for revising and improving our paper, as well as the important guiding significance to our researches. We have studied comments carefully and have made correction which we hope meet with approval. Revised portion are marked in green in the paper. The main corrections in the paper and the responds to the reviewer's comments are as flowing:

Reviewer 4：

1.What kind of inversion did you do for the original CSAMT data? MT or Controlled source inversion including the transition zone effect? It the MT only, it is not a verification. If you use Dipole1D, please do full controlled source inversion to verify your results after NN transformation.

Response:Thank you for your comments. In the comparison of the measured data, we performed the inversion of the CSAMT raw data after obtaining the full domain apparent resistivity, the reason for this is that the geological structure of the work area where this measured data is located is unusually complex, and the data has strong three-dimensional characteristics, so the direct inversion with the source is very ineffective. We used Dipole1D in the CSAMT forward, but the inversion did not use Dipole1D. After a long time of research, we did not find the NN transformation, sorry for that.

2.Why do you do the correction of the near (transition zone) effect and not the true CSAMT inversion (even 1D).

Response:Thank you for your comments, there are various methods to reduce the impact of CSAMT field source effects, obviously direct CSAMT band source inversion is an effective solution, and also includes correction methods based on the definition of region-wide apparent resistivity, etc., and the correction method discussed in this paper is one of them. In addition, after correcting for near-source (overbanding) effects, the csamt data can then be processed using the inversion method of mt, which is simpler and more mature in comparison.

3.Why do you analyze the Kf-Kn results if all your early tests shown that this net give bad results?

Response:Thank you for your comments. In the previous review comments, some reviewers mentioned that they would like us to make a comprehensive comparison between the method of this paper and the traditional method, so we not only tested the results of Kf-Kn in the theoretical model, but also made comparative analyses in the measured data, in order to obtain a more comprehensive comparison.

Journal Requirements:

Response:Thanks for the heads up.The problem with the reference list has been noted, and the formatting has been modified using the ‘Vancouver’ style as required by the journal, with the changes highlighted in green to ensure that the reference list is complete and correct. We have also checked that we have not used withdrawn papers.

---

## [Editor Report · Decision Letter 3]

1 Aug 2024

Deep learning-based near-field effect correction method for controlled source electromagnetic method and application

PONE-D-24-01179R3

Dear Dr. Chen,

We’re pleased to inform you that your manuscript has been judged scientifically suitable for publication and will be formally accepted for publication once it meets all outstanding technical requirements.

Kind regards,

Jinran Wu, PhD

Academic Editor

PLOS ONE

---

## [Editor Report · Acceptance letter]

5 Aug 2024

PONE-D-24-01179R3 

PLOS ONE

Dear Dr. Chen, 

I'm pleased to inform you that your manuscript has been deemed suitable for publication in PLOS ONE. Congratulations! Your manuscript is now being handed over to our production team.

Kind regards, 

on behalf of

Dr. Jinran Wu 

Academic Editor

PLOS ONE